# Efficacy of Transcutaneous Spinal Stimulation versus Whole Body Vibration for Spasticity Reduction in Persons with Spinal Cord Injury

**DOI:** 10.3390/jcm10153267

**Published:** 2021-07-24

**Authors:** Evan B. Sandler, Kyle Condon, Edelle C. Field-Fote

**Affiliations:** 1Shepherd Center, Crawford Research Institute, Atlanta, GA 30309, USA; evan.sandler@shepherd.org (E.B.S.); kyle.condon@shepherd.org (K.C.); 2Program in Biological Sciences, Georgia Institute of Technology, Atlanta, GA 30332, USA; 3Division of Physical Therapy, Emory University School of Medicine, Atlanta, GA 30322, USA

**Keywords:** antispasmodic, electrical stimulation, neuromodulation, paraplegia, pendulum test, tetraplegia

## Abstract

Transcutaneous spinal stimulation (TSS) and whole-body vibration (WBV) each have a robust ability to activate spinal afferents. Both forms of stimulation have been shown to influence spasticity in persons with spinal cord injury (SCI), and may be viable non-pharmacological approaches to spasticity management. In thirty-two individuals with motor-incomplete SCI, we used a randomized crossover design to compare single-session effects of TSS versus WBV on quadriceps spasticity, as measured by the pendulum test. TSS (50 Hz, 400 μs, 15 min) was delivered in supine through a cathode placed over the thoracic spine (T11-T12) and an anode over the abdomen. WBV (50 Hz; eight 45-s bouts) was delivered with the participants standing on a vibration platform. Pendulum test first swing excursion (FSE) was measured at baseline, immediately post-intervention, and 15 and 45 min post-intervention. In the whole-group analysis, there were no between- or within-group differences of TSS and WBV in the change from baseline FSE to any post-intervention timepoints. Significant correlations between baseline FSE and change in FSE were associated with TSS at all timepoints. In the subgroup analysis, participants with more pronounced spasticity showed significant decreases in spasticity immediately post-TSS and 45 min post-TSS. TSS and WBV are feasible physical therapeutic interventions for the reduction of spasticity, with persistent effects.

## 1. Introduction

At discharge from inpatient rehabilitation after spinal cord injury (SCI), more than half of all individuals report experiencing spasticity; a large proportion continue to report that spasticity interferes with function 5 years post-injury [1]. Comprehensively described as “disordered sensori-motor control presenting as intermittent or sustained involuntary activation of muscles,” [2] spasticity impacts the ability to perform functional movements such as transfers, and can lead to contractures and pain [3]. Spasticity is often difficult to manage, and while antispasmodics are the most common management approach, the evidence supporting their value is weak [4]. Moreover, in a survey that acquired responses from 1076 individuals with SCI, only 38% reported that their spasticity was improved by prescribed antispasmodics [5]. By comparison, physical therapeutic interventions such as stretching and exercise were reported to improve spasticity in 48% and 45% of respondents, respectively. In this survey study, spasticity was defined for the respondents in an inclusive way, to encompass characteristics associated with the experience of spasticity, including involuntary spasms, spasms triggered by stimuli, and stiffness.

The common element among all physical therapeutic interventions directed at reducing spasticity is that they activate sensory afferents. Afferent input activates inhibitory spinal interneurons [6,7,8,9,10], and this effect likely underlies the reduction in spasticity in persons with SCI, associated with various forms of afferent stimulation [9,11,12,13,14]. Transcutaneous spinal stimulation (TSS) and whole-body vibration (WBV) are among the approaches that appear to have value for spasticity management [7,15,16,17]; both TSS and WBV activate large-diameter afferent fibers [18,19,20]. Evidence suggests that both electrical (TSS) and mechanical (vibration) approaches to the activation of large-diameter afferents have neuromodulatory effects arising from the activation of inhibitory mechanisms, which likely underlie the observed reduction in spasticity [7,9,12,13,14,15,16,17,21].

Beyond their influence on inhibitory mechanisms, TSS and WBV are well-suited to therapeutic applications because they have modifiable dosing parameters. Recent studies show that 50 Hz TSS, administered for 30 min over the thoracic spine at an intensity below motor threshold, reduces quadriceps spasticity, with effects persisting for up to 2 h post-intervention [7,15,22]. WBV parameters of higher frequency (50 Hz) and longer duration reduced quadriceps up to 45 min post-intervention [16]. Similar to TSS, WBV has been demonstrated to reduce the excitation of the Ia reflex arc through the activation of inhibitory interneurons [20,23]. However, unlike the direct electrical activation of dorsal roots by TSS, WBV repeatedly activates muscle spindles, which provide Ia afferent input to the spinal cord. The more direct effect of TSS on spinal circuits may have a greater impact on the reduction in spasticity.

Clinical feasibility and ease of use are strong determinants of the utility of an intervention in physical rehabilitation and/or the home environment. WBV can present economic limitations, and treatments require an individual to stand on the vibration platform, which is not possible for some who are affected by spasticity. These factors could limit the utility of WBV in the home setting compared to TSS’ ease of electrode application, and the potential for lower-cost stimulation units. TSS may, therefore, be able to benefit a larger group of individuals experiencing lower extremity spasticity. Although both TSS and WBV have been studied separately, no studies have directly compared the effects of TSS and WBV on spasticity. Therefore, our primary aim was to compare the effects of these two approaches on quadriceps spasticity.

## 2. Materials and Methods

This study was conducted with ethical approval from the Shepherd Center Research Review Committee. All participants gave their written informed consent prior to study enrollment, in accordance with the Declaration of Helsinki. This study was registered with clinicaltrials.gov (accessed on 7 October 2014) (NCT02340910).

### 2.1. Subjects

Individuals with SCI were eligible for participation if they met the following inclusion criteria: injury level at or above T12, self-report of at least mild spasticity affecting the lower extremity muscles, ability to stand for ≥1 min using upper extremities for balance only, ability to take a step with at least one leg with or without an assistive device, and ability to rise to a standing position, requiring no more than moderate assistance from one person. Individuals with the following exclusion criteria were not considered for participation: current orthopedic problems preventing participation, history of cardiac irregularity, or progressive or potentially progressive spinal lesions.

### 2.2. Study Design

This study was a supplemental investigation of TSS, incorporated into a single-blind, randomized clinical trial comparing the dose–response effects of WBV on spasticity in individuals with chronic (≥6 months) motor-incomplete SCI. Participants received a single session of TSS within the schema of the WBV dose/frequency randomization. Complete details of the methods of the larger study are described in a prior publication [16]. In this analysis, we compare the effects of TSS with the effects of the WBV dose that had the largest effect on spasticity. Sessions were scheduled at least 1 week apart to minimize the possibility of carryover effects. Within a session, testing was performed prior to the intervention, immediately post-intervention, and 15 and 45 min post-intervention.

### 2.3. Intervention

#### 2.3.1. TSS

To administer TSS, one 5 cm round self-adhesive electrode (cathode) was placed on the lower back over the T11-T12 spinous interspace. One large (10 × 15 cm) self-adhesive butterfly electrode (anode) was placed on the abdomen, over the umbilicus. Tonic stimulation (EMPI Continuum, EMPI, Inc., Clear Lake, SD, USA) was applied using a charge-balanced, biphasic waveform with a pulse width of 400 μs at 50 Hz for 15 min with participants supine. Stimulation intensity was adjusted to the level that evoked paresthesia in the legs, without visible muscle contraction. Stimulation intensity was increased slowly to allow for comfortable adjustment.

#### 2.3.2. WBV

Participants began the WBV session seated on the edge of an adjustable height mat, with feet resting on the WBV platform (Power Plate Pro5, Performance Health Systems, LLC, Northbrook, IL, USA). The mat height was adjusted to allow the participant to rise to standing and return to sitting with minimal effort. The participant rose to stand on the vibration platform with knees slightly flexed (~30° from full extension). Eight cycles of WBV were delivered in 45-s bouts, with 1 min of seated rest between bouts, as previously described [16].

### 2.4. Spasticity Measurement

To evaluate the timecourse of the effects of TSS and WBV on spasticity, spasticity was assessed four times during each session: prior to the start of the intervention (baseline), immediately after the conclusion of the intervention (immediate), 15 min after the intervention (15-min post-intervention), and 45 min after the conclusion of the intervention (45-min post-intervention). We tested the leg the participant reported to be most spastic at the time of study enrollment, and the same leg was tested in each session.

The pendulum test was used to assess stretch-induced quadriceps spasticity. Participants were positioned semi-reclined with the test leg flexed at the knee, lower leg pendant over the edge of the mat, and shoe removed. An electrogoniometer (SG150, Biometrics Ltd., Newport, UK) was affixed to the test leg with the arms aligned with the thigh and shank, and the axis aligned with the knee joint center, to record knee angle during the pendulum test. The non-test leg was supported on a padded chair with the knee extended. Grasping the heel of the test leg, the examiner extended the knee and held the leg in this position for 30 s to allow movement-related excitability to dissipate. The examiner then released the heel, allowing the test leg to drop. The first swing excursion (FSE), the angle at which the swinging knee first reversed from flexion to extension in response to the reflexive contraction of the quadriceps, was the primary measure of spasticity, wherein a larger angle indicates less spasticity [24]. Comparisons have shown FSE to be the best measure of quadriceps spasticity relative to other outcomes of the pendulum test, including the relaxation index and number of oscillations [25]. Acquisition and analysis of FSE was conducted using Spike software (Cambridge Electronic Design Limited, Cambridge, England). The average FSE of 3 trials of each test session was used for analysis.

### 2.5. Data Analysis

All data analyses were performed in SPSS version 27 (IBM, London, UK). Data are presented as the mean ± SD. All analyses were completed for the entire sample, and for high spasticity and low spasticity subgroups, to determine the effect of spasticity severity on responsiveness to intervention. Participants were grouped into high- and low-spasticity subgroups, based on the previously published median baseline FSE (46.6°) of this study sample, which showed differential effects of WBV based on baseline spasticity [16]. The subgroups were: high spasticity = baseline FSE < 46.6° and low spasticity = baseline FSE > 46.6°. Subgrouping was determined by the FSE of the baseline test for the TSS and WBV sessions individually, as some participants who met the criteria for high spasticity during one session did not meet that criteria for the other session.

Effect sizes were calculated using Cohen’s *d* based on the pooled variance of the compared values. Effect sizes were categorized as small (0.08), moderate (0.31), or large (0.55) based on the recommendations of a recent meta-analysis of rehabilitation research outcomes [26]. For significance testing α = 0.05 was considered significant. Paired *t*-tests were used to test for differences between baseline FSE of the two interventions, and to identify between-intervention differences in change scores in FSE at each timepoint. Independent *t*-tests were used to determine differences in the change from baseline between interventions. Paired *t*-tests were used to identify within-condition effects of TSS, comparing baseline FSE to each post-intervention measurement (immediate, 15-min post-intervention, and 45-min post-intervention). Pearson correlations were calculated to determine the relationship of change scores between interventions. Within each intervention, Pearson correlations were calculated to determine the relationship between spasticity severity, as measured by baseline FSE, and responsiveness to each intervention.

## 3. Results

Thirty-two participants completed both the TSS intervention and the high-frequency/long-duration WBV intervention. Participants were aged from 23 to 65 years old, and included 26 men and 6 women. Of the 32 participants, 9 were classified as American Spinal Injury Association Impairment Scale (AIS) C and 23 were classified as AIS D. Among the participants, 15 used no antispasmodic medications, 11 used baclofen only, 1 used gabapentin only, 3 used two medications (baclofen plus either gabapentin or tizanidine), and 2 used three medications (baclofen, gabapentin, and dantrolene). For the TSS intervention session, 18 participants met the high-spasticity subgroup criteria. For the WBV intervention, 13 participants met the high-spasticity subgroup criteria. Detailed demographic information, including pharmacological use by participants, is available elsewhere [16].

### 3.1. Effect of TSS on Quadriceps Spasticity

Mean baseline FSE for all participants during the TSS session was 44.8° ± 16.7°. Analysis of the full sample showed no overall effect of TSS. Baseline FSE was not different from any of the post-intervention assessments, including baseline vs. immediate post-intervention (*p* = 0.42), baseline vs. 15-min post-intervention (*p* = 0.58), and baseline vs. 45-min post-intervention (*p* = 0.50) (Table 1).

Mean baseline FSE for the high-spasticity subgroup was 32.7° ± 9.50°, and for the low-spasticity subgroup, mean baseline was 60.3° ± 9.17°. Upon stratification into high-spasticity and low-spasticity subgroups, the differences between baseline FSE and post-intervention assessments were identified only in the high-spasticity subgroup. In the high-spasticity subgroup, the effect size for difference between FSE at baseline and immediately post-intervention was moderate (*d* = 0.48) and statistically significant (*p* = 0.036). The difference between FSE at baseline and 15-min post-intervention has a moderate effect size (*d* = 0.32), and this difference approached statistical significance (*p* = 0.100). The effect size for difference between FSE at baseline and 45-min post-intervention was large (*d* = 0.55) and statistically significant (*p* = 0.035).

### 3.2. Effect of WBV on Quadriceps Spasticity

As noted earlier, outcomes of the high-frequency/long-duration WBV session have been previously reported, and are included here to allow for comparison with TSS outcomes [16]. Mean baseline FSE for all participants during the high-frequency/long-duration WBV session was 50.34° ± 20.8°. Analysis of the full sample showed no overall effect of WBV. Baseline FSE was no different from any of the post-intervention assessments (*p* > 0.05 for each pairwise comparison; Table 1).

Mean baseline FSE for the high spasticity subgroup was 29.4° ± 8.0° and for the low -spasticity subgroup, mean baseline was 64.6° ± 13.1°. Upon stratification into high-spasticity and low-spasticity subgroups, differences between baseline FSE and post-intervention assessments were identified only in the high-spasticity subgroup. In the high-spasticity subgroup, the effect size for difference between FSE at baseline and immediately post-intervention was moderate (*d* = 0.33) but not statistically significant (*p* > 0.05). The effect size for difference between FSE at baseline and 15-min post-intervention was large (*d* = 0.90) and statistically significant (*p* = 0.014). The difference between FSE at baseline and 45-min post-intervention showed a large effect size (*d* = 0.73), but was not statistically significant (*p* > 0.05). In the low-spasticity subgroup, effect sizes were moderate and significant for difference between FSE at baseline and 15- and 45-min post-intervention (*d* = −0.46, *d* = −0.44).

### 3.3. Differences between TSS and WBV

Analyzing all participants, no significant differences between the TSS and WBV were observed in change in FSE at any timepoint, including baseline vs. immediate (*p* = 0.63), baseline vs. 15-min post-intervention (*p* = 0.50), or baseline vs. 45-min post-intervention (*p* = 0.48) (Figure 1). Significant differences were, however, observed in baseline FSE between the TSS and WBV conditions (*p* = 0.034). After subgrouping individuals in accordance with baseline FSE, participants who were in the same subgroup at baseline testing for both TSS and WBV were included in between-intervention subgroup analyses. Baseline FSE was not significantly different between conditions after subgrouping (high: *p* = 0.64, low: *p* = 0.12). When analyzing subgroup change in FSE, no significant differences between TSS vs. WBV were observed at any timepoint, including baseline vs. immediate post-intervention (*p* = 0.61, *p* = 0.69), 15-min post-intervention (*p* = 0.34, *p* = 0.45), or 45-min post-intervention (*p* = 0.80, *p* = 0.98), for the high- and low-spasticity subgroups, respectively.

### 3.4. Influence of Baseline Spasticity on Change in Quadriceps Spasticity

Correlations between baseline FSE and change in FSE were significant at all timepoints for the TSS intervention (immediate: *r* = −0.44, *p* = 0.012, 15-min post-intervention: *r* = −0.36, *p* = 0.04) (Figure 2). Correlations were significant for the WBV intervention only at 15-min and 45-min post-intervention timepoints (15-min post-intervention: *r* = −0.55, *p* = 0.001). The strongest correlations for both intervention conditions were observed 45-min post-intervention (TSS: *r* = −0.49, *p* = 0.005, WBV: *r* = −0.57, *p* = <0.001), with the negative correlations indicating that those with high spasticity (smaller FSE) had a greater change.

## 4. Discussion

This study compared the single-session effects on quadriceps spasticity of TSS and WBV, two robust forms of afferent stimulation. When all participants were considered together, neither the TSS nor the WBV intervention demonstrated significant differences between baseline and any post-intervention timepoint. Subgrouping participants based on baseline FSE demonstrated the influence of baseline spasticity on responsiveness to TSS, as was reported with WBV [16]. Participants with low spasticity (>46.6° baseline FSE) did not demonstrate significant differences between baseline FSE and any post-intervention timepoint. However, in the high-spasticity subgroup (<46.6° baseline FSE), TSS was associated with an increase in FSE at all post-intervention timepoints, indicating a decrease in quadriceps spasticity persisting for at least 45 min post-intervention. The greatest differences in FSE were found immediately and at 45-min post-intervention. Moreover, the change in FSE immediately post-intervention was associated with a moderate effect size, with a large effect size at 45-min post-intervention.

No differences between TSS and WBV regarding their effect on spasticity were identified at any post-intervention timepoint. The lack of difference between TSS and WBV was true for both the high-spasticity and low-spasticity subgroups, indicating that the two interventions have equivalent effects. Importantly, baseline FSE was significantly correlated with changes in FSE at all timepoints, for both the TSS and WBV interventions. This relationship indicates that participants with higher spasticity demonstrated a greater reduction in spasticity with each of the interventions, compared to those with lower spasticity.

As previously reported, WBV at 50 Hz for eight 45-s bouts resulted in a reduction in quadriceps spasticity in participants with high spasticity at 15-min and 45-min post-intervention assessment timepoints [16]. The differences between baseline FSE and at the 15-min and 45-min post-intervention assessments were associated with large effect sizes. This delayed antispasmodic effect of WBV is consistent with prior reports [15,17]. While no differences were observed between the TSS and WBV interventions, in the TSS condition, the high-spasticity subgroup demonstrated a significant and immediate reduction in spasticity, persisting at the 15- and 45-min post-intervention assessments. Conversely, in the WBV condition, the high-spasticity subgroup did not exhibit a significant reduction in spasticity at the immediate post-intervention assessment. However, a more robust change, as measured by effect size, was observed after WBV as compared to TSS at the 45-min post-intervention assessment. For both interventions, in contrast to the improvement in the high-spasticity subgroups, the low-spasticity subgroup demonstrated increased spasticity after intervention.

Although both TSS and WBV preferentially activate large-diameter afferent fibers [18,19,20], mechanical and electrical forms of stimulation may be associated with different neuromodulatory effects that account for the different effect sizes observed between interventions. While the same presynaptic inhibitory effects attributed to reductions in spasticity after vibration have been suggested in studies of TSS based on reflex testing [7,22,27], vibration also has excitatory influences on spinal circuits [28]. Vibration at 50 Hz has been shown to increase muscle activation in healthy subjects [29]. In addition, in a study of the multi-session effects of WBV on quadriceps spasticity, within-session testing identified an increase in spasticity immediately post-intervention, followed by reduced spasticity 15-min post-intervention [17]. The balance between the excitatory and inhibitory mechanisms, or the “vibration paradox”, was postulated to account for the delayed spasticity reduction [17]. Prior studies have suggested that the activation of spinal circuits by TSS parallels that of epidural stimulation, with frequency-dependent effects [30,31]. While the frequency of WBV (50 Hz) was the same as that of TSS, there is likely some damping that occurs with WBV, which may result in an effective activation frequency on neural structures that is less than 50 Hz.

Unlike WBV, electrical stimulation can activate smaller diameter afferents including Aδ and C fibers in addition to Ia fibers (dependent upon stimulation intensity) [32]. Moreover, an increase in gain of the muscle spindle responses via augmented gamma motoneuron drive has been proposed as a potential mechanism for the hyper-responsiveness to stretch after spinal cord injury [33]. In preclinical studies, nociceptor activation has been shown to increase muscle spindle firing rates associated with increased gamma motoneuron activity [34,35,36]. In humans, however, the induced nociception during a relaxed state and during muscle contraction did not affect muscle spindle discharge rates, confuting the excitatory gamma motoneuron response to nociceptive stimuli observed in preclinical models [35]. The same absence of excitation to the gamma motoneuron exists in individuals with chronic spinal cord injury; therefore, it is unlikely that the activation of nociceptor fibers is responsible for the differences seen between TSS and WBV [37].

TSS and WBV target different spinal segments during stimulation. The T11-T12 interspace placement of the cathode during TSS targets the rostral portion of the lumbar enlargement, highly favoring motoneuron pools of the quadriceps [38,39]. Although the quadriceps are targeted when standing on the WBV platform with a slight bend at the knee, the muscle afferents of all lower extremity musculature are active during vibration, as evidenced by soleus reflex modulation in non-injured individuals and those with SCI [20,28,40]. Neuromodulatory effects, as elucidated by paired-pulse stimulation studies, have demonstrated the potential for inhibition by heteronymous circuits [38,41,42,43], providing evidence of intersegmental modulation. Moreover, the voluntary contraction of non-tested musculature during spinal reflex testing has also demonstrated the influence of intersegmental circuits on reflex modulation [44]. The amount of excitation or inhibition from other muscles onto the quadriceps may influence its responsiveness to stretch post-vibration, accounting for the differential effects of TSS and WBV.

The change in responsiveness to stretch in persons with higher levels of spasticity observed in the present study adds to the growing body of literature of the beneficial effects of physical therapeutic interventions on spasticity in persons with SCI. In a comparative study of physical therapeutic interventions, TSS, stretching, and continuous passive movement all demonstrated immediate and persistent reductions in spasticity in persons with chronic SCI [15], congruent with the results found in this study. A reduction in spasticity in persons with SCI has been observed after other forms of electrical stimulation, including transcutaneous electrical nerve stimulation [14] and neuromuscular electrical stimulation/functional electrical stimulation [45].

Physical therapeutic interventions such as TSS and WBV are clinically accessible and avoid the negative side effects associated with pharmacological interventions for spasticity management, such as fatigue, sleepiness [46] and muscle weakness [47]. It is worth noting that antispasmodic medications are prescribed to be taken two or three times daily, as the persistence of their effects is limited. Likewise, it is possible that it is necessary to administer physical therapeutic interventions several times a day to achieve optimal spasticity management. In this case, the greater portability of TSS compared to WBV, and its potential to be “wearable”, may make it the more advantageous approach.

### Limitations

One limitation of this work relates to the within-participant variability in the quadriceps spasticity measurement. When comparing TSS and WBV based on spasticity at baseline, it was necessary to exclude individuals who did not demonstrate consistently high or low spasticity at baseline for both interventions from the analysis. As demonstrated in our previous work, an individual’s quadriceps spasticity may vary day-to-day. By excluding the nine individuals whose baseline spasticity did not result in the same high versus low spasticity subgroup for both conditions, we were able to compare responses in the same individuals across interventions. However, this resulted in a reduced sample size that may have decreased the power to detect differences between interventions. Another limitation of this study is the use of a single set of parameters to achieve antispasmodic effects. While studies of epidural stimulation have been used as guides for parameter selection, systematic studies are needed to explore the optimal parameters, frequency and intensity, for spasticity reduction. Lastly, comparisons between and within interventions can only be drawn up to 45-min post-intervention. The persistence of effects beyond 45 min should be explored and may prove different between interventions.

## 5. Conclusions

In persons with chronic motor-incomplete SCI with higher levels of quadriceps spasticity, transcutaneous spinal stimulation and whole-body vibration are associated with reduction in spasticity. Both interventions provide persistent effects for at least 45 min after cessation of the intervention. While this is a single-session comparison study, it is necessary to determine the potential of an intervention when treating this participant population. With the limited financial resources allocated for therapeutic intervention throughout the recovery period, future research should focus on identifying whether an individual with SCI is a potential responder prior to performing an intervention.

## Figures and Tables

**Figure 1 jcm-10-03267-f001:**
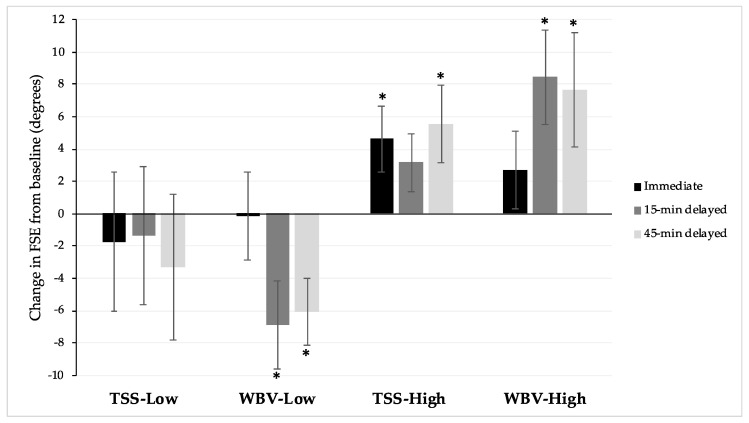
Mean change in FSE from baseline by subgroup. TSS, Transcutaneous spinal stimulation; WBV, whole-body vibration. Immediate (black bars), 15-min post-intervention (dark grey bars), and 45-min post-intervention (light grey bars) results are presented for participants with high (baseline FSE < 46.6°) and low (baseline FSE > 46.6°) spasticity. Asterisks (*) denote a significant (*p* < 0.05) change from baseline FSE.

**Figure 2 jcm-10-03267-f002:**
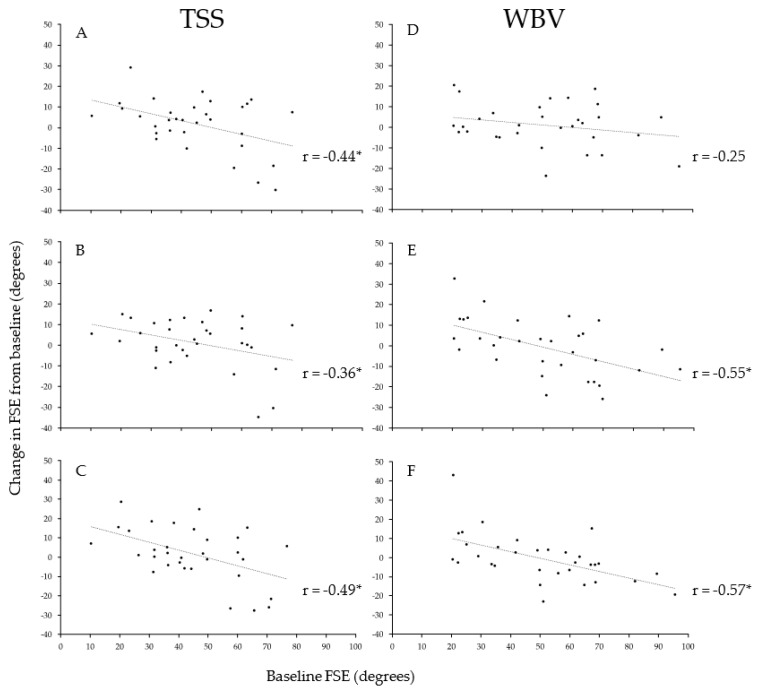
Correlation of baseline spasticity and change. Change in quadriceps spasticity as a function of individuals’ baseline spasticity. Correlations of baseline FSE to immediate (**A**,**D**), 15-min post-intervention (**B**,**E**), and 45-min post-intervention (**C**,**F**) timepoints are presented for the whole group. * *p* < 0.05.

**Table 1 jcm-10-03267-t001:** Mean first swing excursion (FSE) by group.

	**Whole Group FSE**			
	**Baseline**	**Immediate**	**15-min Post-Intervention**	**45-min Post-Intervention**
TSS (*n* = 32)	44.78 ± 16.70	46.64 ± 15.91 (0.11)	45.97 ± 16.65 (0.07)	46.46 ± 15.72 (0.10)
WBV (*n* = 32)	50.34 ± 20.75	51.96 ± 20.85 (0.05)	49.69 ± 17.48 (−0.03)	49.86 ± 17.10 (−0.03)
	**High Spasticity Group FSE**			
	**Baseline**	**Immediate**	**15-min Post-Intervention**	**45-min Post-Intervention**
TSS (*n* = 18)	32.69 ± 9.50	37.33 ± 9.80 * (0.48)	35.85 ± 10.31 (0.32)	38.24 ± 10.60 * (0.55)
WBV (*n* = 13)	29.36 ± 8.03	32.10 ± 8.51 (0.33)	37.91 ± 10.81 * (0.90)	37.13 ± 12.60 (0.74)
	**Low Spasticity Group FSE**			
	**Baseline**	**Immediate**	**15-min Post-Intervention**	**45-min Post-Intervention**
TSS (*n* = 14)	60.33 ± 9.17	58.60 ± 14.28 (−0.14)	58.97 ± 14.09 (−0.11)	57.04 ± 15.11 (−0.26)
WBV (*n* = 19)	64.62 ± 13.10	64.51 ± 15.85 (−0.01)	57.76 ± 16.71 * (−0.46)	58.58 ± 14.15 * (-0.44)

Results represent means ± SD. Asterisks (*) denote significant (*p* < 0.05) difference from baseline mean FSE. Effect sizes for within-condition pre- and posttest comparisons listed in parentheses. TSS, Transcutaneous spinal stimulation; WBV, whole-body vibration.

## Data Availability

The data presented in this study are available upon request from the corresponding author.

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
