# Peer review of "Efficacy of Transcutaneous Spinal Stimulation versus Whole Body Vibration for Spasticity Reduction in Persons with Spinal Cord Injury"

_jcm, 2021, doi:10.3390/jcm10153267_

Round 1

Reviewer 1 Report

This is a timely and high-quality study comparing the effect of tSCS and WBV on spasticity control in people with SCI. I have only a few comments to make.

Abstract:

Please include the polarity of electrode (anode) placed on the T11/T12 area.

Mention that there are no pre-post differences with either tSCS or WBV (just for clarity).

Mention the posture of the person with tSCS (supine?).

Introduction:

Line 33: Would be good to justify use of PT in this study, in context with involuntary spasms.

Materials and Methods:

Line 101: mention where anode and cathode are placed.

Line 129: justify why first swing excursion was used as the outcome parametric, rather than R2n.

Line 157: mention if a Bonferroni adjustment was made.

Results:

Line 160: A description of demographic and clinical characteristics of the cohort would be useful, even though some data has been published previously.

Line 195: Include the observation of a decrease FSE in the WBV-Low group, especially with a good effect size shown in Table 1.

Line 224 Fig 1: Replace “delayed” by “follow-up”.

Line 230: Would be useful to include a guideline to dived between low and high spasticity at 46º.

Discussion

Line 242: reference to a decrease in FSE in the low spasticity group with WBV, especially as this is an interesting observation.

Line 298: include information about cathode in the methods section.

Line 332: typo with “higho”

Reviewer 2 Report

Overall, I find this manuscript to be well written and clinically significant.  I have only a few comments/critiques:

My primary critique concerns the lack of information on the participants/methods. In particular, it is critical to include information about medications; whether any/all of of these participants were taking any antispasmodics, and if so, which and at what dose? It is not possible to form many sound conclusions regarding a spasticity study without this information. It would be of interest to know what the injury levels and time since injury were, and whether this had any influence on the results. A table with this information as well as the effects of the interventions per individual would greatly improve this manuscript. If this data is unavailable, it should be noted as a major limitation. Regarding the methods, were the individuals seated during TSS, or standing as with WBV? this information should be included as well.

Second, some of the significant limitations were not mentioned: 1) only one set of parameters for both WBV and TSS were used- although evidence was appropriately cited for the selection of these parameters, it is entirely possible that they were/are not the optimal parameters for spasticity reduction. For that matter, data from epidural and transcutaneous studies could be taken to suggest that there is no optimal parameter for all subjects, rather the parameters considered 'optimal' may be unique for each individual, based on the unique aspects of their injury and subsequent adaptive or maladaptive neuroplastic changes. TSS is also routinely applied at multiple locations, which might also produce different results. 2) when looking at ESS and TSS data in particular, it seems possible that there may be chronic effects on spasticity, or effects on spasticity when stimulation is combined with weightbearing activity, which may or may not be present during a single session. This possibility/limitation also deserves mention.

Third, this data appears to suggest that these interventions may actually increase spasticity in those with 'mild spasticity'. Again, it is hard to evaluate this aspect partly because it is unknown to the reader whether these individuals truly had less spasticity, or whether their spasticity was being better managed pharmacologically while others were not on antispasmodics. Regardless, no discussion of this point is included in the current manuscript despite its obvious clinical significance. 

minor comments:

There were a few instances of grammatical errors: i noted an error on line 93, and an incomplete sentence on lines 95-96.  

line 300 makes reference (#37) to:

Sayenko DG, Atkinson DA, Floyd TC, Gorodnichev RM, Moshonkina TR, Harkema SJ, Edgerton VR, Gerasimenko YP. Effects of paired transcutaneous electrical stimulation delivered at single and dual sites over lumbosacral spinal cord. Neurosci Lett. 2015 Nov 16;609:229-34. doi: 10.1016/j.neulet.2015.10.005. Epub 2015 Nov 4. PMID: 26453766; PMCID: PMC4679579.

I believe the more correct reference here would be:

Sayenko DG, Atkinson DA, Dy CJ, Gurley KM, Smith VL, Angeli C, Harkema SJ, Edgerton VR, Gerasimenko YP. Spinal segment-specific transcutaneous stimulation differentially shapes activation pattern among motor pools in humans. J Appl Physiol (1985). 2015 Jun 1;118(11):1364-74. doi: 10.1152/japplphysiol.01128.2014. Epub 2015 Mar 26. PMID: 25814642; PMCID: PMC4451290.

lines 298-310  it is not accurate to suggest that WBV affects all LE afferent while TSS does not. Multiple studies you have cited demonstrate that TSS at T11-T12 can most certainly affect lower limb muscles, even those innervated by the S1-S2 segments.
